# Comparing Fungal Sensitivity to Isothiocyanate Products on Different *Botrytis* spp.

**DOI:** 10.3390/plants13060756

**Published:** 2024-03-07

**Authors:** Víctor Coca-Ruiz, Josefina Aleu, Isidro G. Collado

**Affiliations:** 1Departamento de Química Orgánica, Facultad de Ciencias, Universidad de Cádiz, 11510 Cádiz, Spain; victor.coca@uca.es; 2Instituto de Investigación en Biomoléculas (INBIO), Universidad de Cádiz, 11510 Cádiz, Spain

**Keywords:** fungi, *Botrytis*, pathogenicity, glucosinolates

## Abstract

Glucosinolates, the main secondary metabolites accumulated in cruciferous flora, have a major impact on fortifying plant immunity against diverse pathogens. Although *Botrytis cinerea* exhibits varying sensitivity to these compounds, current research has yet to fully understand the intricate mechanisms governing its response to glucosinolates. Different species of the genus *Botrytis* were exposed to glucosinolate-derived isothiocyanates, revealing that *B. fabae*, *B. deweyae*, and *B. convolute*, species with the *mfsG* transporter gene (Bcin06g00026) not detected with PCR, were more sensitive to isothiocyanates than *Botrytis* species containing that gene, such as *B. cinerea*, *B. pseudocinerea*, and *B. byssoidea*. This finding was further corroborated by the inability of species with the *mfsG* gene not detected with PCR to infect plants with a high concentration of glucosinolate-derived isothiocyanates. These results challenge established correlations, revealing varying aggressiveness on different plant substrates. An expression analysis highlighted the gene’s induction in the presence of isothiocyanate, and a bioinformatic investigation identified homologous genes in other *Botrytis* species. Our study underscored the importance of advanced biotechnology to help understand these proteins and thus offer innovative solutions for agriculture.

## 1. Introduction

Plants require a systemic defense system that enables them to effectively recognize invading molecules in cells damaged by pathogens and immediately activate immune responses [1,2]. Several specialized secondary metabolites that are toxic and repellent to pests and pathogens are produced to overcome pest and pathogen invasions [3,4]. Plant secondary metabolites can be grouped into three chemically distinctive main classes: terpenes, phenolics, and nitrogen-containing compounds [5].

Among them, glucosinolates (GLSs) are a structurally well-defined group of anionic natural products that serve as vital defense compounds in plants belonging to the Brassicales order and include various brassicaceous vegetables such as cabbage, radish, and broccoli [6]. These GLSs undergo hydrolysis facilitated by specific myrosinase enzymes known as thio-β-glucosidases. This hydrolysis process leads to the formation of unstable aglycones, yielding primarily isothiocyanates (ITCs), with the common structure R-N=C=S (Figure 1), nitriles R-CN, or thiocyanates R-S=C=N. In this process, the specific type of GLSs, the local environment, and the presence of specifier proteins have a bearing on the outcome [7,8,9]. The standard product of the reaction is isothiocyanate; the other two compounds mainly occur in the presence of specifier proteins that convert the glucosinolate aglucone to other products [7]. These resulting breakdown products operate as deterrents or toxins against potential aggressors.

Fungal pathogens pose a significant threat to agricultural productivity and food security by targeting a wide array of crops. To survive and thrive in hostile environments, these fungi have evolved sophisticated defense mechanisms against the above-indicated toxic compounds. Among the primary mechanisms employed by fungi to counteract these toxic insults is the active efflux mediated by ATP-binding cassette (ABC) or major facilitator superfamily (MFS) transporters [10,11,12].

*B. cinerea* is the agent causing gray mold disease, which affects a total of 586 genera of vascular plants, accounting for over 1400 ornamental and agriculturally important plant species [13,14,15,16]. This fungus is a typical necrotroph whose infective cycle includes the destruction of plant cells and the maceration of plant tissue. This notorious gray mold pathogen inflicts substantial economic losses both during crop growth and the postharvest stages, necessitating intensive chemical and biological control measures [17,18]. However, the rampant use of fungicides has fueled the emergence of fungicide-resistant strains, exacerbating the challenge of disease management [19,20].

Deciphering the mechanisms underlying *B. cinerea*’s ability to evade host defenses is paramount for developing effective disease management strategies. Recent investigations have revealed considerable variability in *B. cinerea*’s susceptibility to glucosinolates (GLSs) and their breakdown products, with *Alternaria brassicicola*, a specialized Brassica pathogen, demonstrating heightened tolerance to GLSs and their breakdown components [21,22]. It is worth highlighting the work of Vela-Corcía [23], in which deletion of the *mfsG* gene in *B. cinerea* resulted in decreased tolerance to ITCs in vitro and reduced virulence toward *Arabidopsis thaliana*, underscoring the importance of this transporter in fungal pathogenicity [24]. In addition, this transporter exhibits differential expression patterns in response to various ITCs, with heightened expression levels observed during interactions with wild-type *A. thaliana* [23]. Furthermore, additional studies have identified other detoxification mechanisms, such as the one present in *Sclerotinia sclerotiorum*, which effectively decomposes ITCs via enzymatic hydrolysis, with the SsSaxA protein playing a crucial role in it. However, a mutant lacking the *SssaxA* gene was unable to thrive in host plants producing glucosinolates [25]. Moreover, other studies have emphasized the importance of this family of transporters (the MFS) being required, for example, for multidrug/multixenobiotic resistance in yeast (*Saccharomyces cerevisiae*) [26].

Understanding the intricate interplay between fungal transporters and host defense compounds is crucial for devising sustainable strategies to mitigate fungal diseases in agricultural systems. Targeting specific transporters involved in fungal detoxification processes holds promise for the development of novel fungicides with enhanced efficacy and reduced environmental impact [27,28]. Consequently, elucidating the regulatory networks governing transporter expression and activity may provide valuable insights into the evolution of fungicide resistance and guide the design of innovative approaches to combat fungal pathogens [29].

The elucidation of these mechanisms not only enhances our understanding of fungal pathogenesis but also informs the development of strategies for sustainable disease management. So, the aim of this work is to study the *Botrytis* species’ tolerance to one type of glucosinolate breakdown product (isothiocyanates) and assess the evolutionary conservation and functional implications of the *mfsG* gene.

## 2. Results

### 2.1. mfsG Gene Present in Only a Few Botrytis Species

A bioinformatic analysis (blastp and blastn) was performed to detect the presence of homologous genes in other *Botrytis* species. The blastp-blastn analysis showed that the *mfsG* gene was only detected in *B. cinerea*, *B. pseudocinerea*, *B. byssoidea*, *B. fragariae*, and *B. medusae*, as detailed in Table 1. In addition, the homologous genes with the highest identity were those corresponding to the *B. pseudocinerea* and *B. medusae* species with a percentage of identity higher than 90% (Table 1). On the other hand, the *B. fragariae* and *B. byssoidea* species were the ones with the lowest identity, with an identity higher than 80% in both cases (Table 1).

However, an experimental approach was carried out on six *Botrytis* species available in our laboratory for the detection of this gene. The amplification with the primer pairs CKmfsG1/CKmfsG2, CKmfsG3/CKmfsG4, and CKmfsG1/CKmfsG4 (Appendix A) only gave results in the *B. cinerea* B05.10, *B. pseudocinerea* VD165, and *B. byssoidea* MUCL94 species, as can be seen in Figure 2, while it was not detected in the *B. deweyae* B1, *B. convolute* MUCLII595, and *B. fabae* 2220 species.

### 2.2. Amino Acid Conservation of the mfsG Gene

The conservation of amino acids within the catalytic center proposed by Vela-Corcía in 2019 [23] (Trp54, Phe254, Gln344, and Phe369) was assessed using the amino acid sequences from *B. cinerea* B05.10, *B. fragariae* BVB16, and *B. byssoidea* MUCL94 (the only *Botrytis* species in which the existence of this protein was detected in the blastp search). This analysis revealed a notable conservation of these amino acids across the species, suggesting their potential significance in the protein’s activity (Figure 3).

This study also looked into the conservation of the amino acids predicted to bind to the isothiocyanates BITC and PITC (Trp54) and PhITC (Arg290) via hydrogen bonds [23]. The results in Figure 3 show that most of these amino acids in various *Botrytis* species are conserved. 

This comprehensive approach, integrating experimental findings with bioinformatic analyses, contributes to a more nuanced understanding of the molecular landscape across *Botrytis* species. The conservation patterns identified pave the way for future research, providing a basis with which to explore the functional implications of these conserved amino acids and their role in the broader context of plant-pathogen interactions.

### 2.3. Evaluation of the Tolerance of Botrytis Species to Isothiocyanate Products

Six different *Botrytis* species were evaluated for their tolerance to three isothiocyanate products: propyl isothiocyanate (PITC), benzyl isothiocyanate (BITC), and 2-phenethyl isothiocyanate (PhITC) (Figure 1). In three of these six species (*B. cinerea* B05.10, *B. pseudocinerea* VD165, and *B. byssoidea* MUCL94), the *mfsG* gene was detected amplifying different fragments of the gene by PCR, while in the other three species (*B. deweyae* B1, *B. fabae* 2220, and *B. convolute* MUCLII595), the *mfsG* gene was not detected with the amplification of different fragments of the gene by PCR.

The *Botrytis* species in which the *mfsG* transporter gene was detected showed higher tolerance (less growth inhibition percentage) than those species in which it was not detected (Figure 4). Of the species in which the *mfsG* gene was detected, *B. pseudocinerea* VD165 proved to be the most resistant to these compounds, showing the highest IC50 value observed across all the experiments performed for the three isothiocyanates tested (Table 2). Regarding *B. byssoidea* MUCL94, it presented a higher IC50 value than *B. cinerea* B05.10, showing a higher tolerance to PITC than *B. cinerea* B05.10 (Table 2 and Figure 4). However, in the case of BITC and PhITC, *B. cinerea* showed a higher IC50 value than *B. byssoidea* MUCL94, indicating a higher degree of resistance to BITC and PhITC (Table 2 and Figure 4) than *B. cinerea* B05.10. The difference was more accentuated in the case of BITC, with both species showing a more similar IC50 value for PhITC. On the other hand, of the species in which the *mfsG* gene was not detected by PCR, *B. deweyae* was the most tolerant (Figure 4 and Table 2). Of the species most sensitive to the isothiocyanate products, *B. convolute* was slightly more resistant when compared to *B. fabae*. In addition, both *B. fabae* and *B. convolute* showed a lower IC50 value for PITC, BITC, and PhITC than *B. deweyae* (Table 2).

However, Figure 4 shows that the tolerance of all *Botrytis* species to isothiocyanates depends on the concentration of the isothiocyanates. BITC was the most toxic of the compounds (the one that showed the highest IC50 values for all the *Botrytis* species tested; Table 2), followed by PhITC. PITC was the least toxic compound, even failing to completely inhibit growth at the concentrations used on the species with the *mfsG* gene not detected by PCR, such as *B. deweyae*. 

These inhibition data for the *Botrytis* species with no *mfsG* gene detected coincide with those reported by Vela-Corcía in 2019, with an inhibition to these compounds similar to that presented by the *mfsG* knock-out gene in *B. cinerea* B05.10 [23], supporting the idea of the lack of the gene in *B. deweyae* B1, *B. fabae* 2220, and *B. convolute* MUCLII595.

### 2.4. Infection Assays of Botrytis Species with mfsG Gene Detected and Not Detected on Different Brassicaceae Species

Infectivity assays were performed to assess the significance of the resistance conferred by this gene to isothiocyanates, which represent one family of plant defense compounds. Consequently, *Brassica oleracea* and *Raphanus sativus* leaves, *Raphanus sativus* tubers, and *Brassica oleracea* var. italica vegetables were infected (Figure 5). All of these have previously been reported to contain high glucosinolate levels [35]. In addition, previous studies reported that glucosinolate production varies between cultivars [36,37].

In all the samples analyzed, the species in which the *mfsG* gene was detected by PCR were able to infect these substrates, in contrast to the *Botrytis* species with no *mfsG* gene detected by PCR. Surprisingly, this was not the case with broccoli, where just the opposite occurred, i.e., the species with no *mfsG* gene detected were more successful in infecting the substrate than the species containing the *mfsG* gene.

Regarding the infection of cabbage and radish leaves and radish tubers, *B. pseudocinerea* VD165 was the most aggressive species, followed by *B. cinerea* B05.10 and *B. byssoidea* MUCL94, except in the case of radish in tuber form, where *B. cinerea* B05.10 was the most aggressive, followed by *B. pseudocinerea* VD165 and *B. byssoidea* MUCLII595. The species with no *mfsG* gene detected by PCR were unable to infect these substrates. In radish leaves, a small lesion was caused by *B. deweyae* B1 and *B. fabae* 2220 infections (with no significant differences between them), which were unable to develop further. This was also the case for the *B. deweyae* B1 infection in radish tubers, where the lesion remained small. As in the previous case, the infection of these species by *B. pseudocinerea* VD165 and *B. byssoidea* MUCL94 has been described in vitro for the first time (Figure 5). 

### 2.5. Isothiocyanate Products and Infection Processes Induce mfsG Gene Expression

An expression analysis revealed that all three evaluated isothiocyanates (PITC, BITC, and PhITC) induced the expression of the *mfsG* gene in *B. cinerea* B05.10, *B. pseudocinerea* VD165, and *B. byssoidea* MUCL94 (Figure 6). Furthermore, each compound induced expression in each species at the same level, with no significant differences between them. However, PITC was the only isothiocyanate that showed lower gene induction in the three *Botrytis* species studied compared to BITC and PhITC, although there were no significant differences between the latter two (Figure 6).

On the other hand, the expression evaluation during infection on different plant substrates of the Brassicaceae family revealed gene expression of the *mfsG* gene in all the *Botrytis* species studied (Figure 6). It is noteworthy that a lower induction was observed during infection compared to the evaluation of pure compounds. These results make sense because the two studies are not comparable due to the variability in glucosinolate composition among plant species, as there are other types of glucosinolates besides isothiocyanates that may influence the expression levels of the *mfsG* gene.

On the other hand, contrary to what was observed in the *mfsG* expression of pure isothiocyanates, in the case of *Brassica oleracea* and *Raphanus sativus* leaves, induction was not uniform across the three *Botrytis* species (Figure 6). During the *Brassica oleracea* infection, higher induction of the *mfsG* gene was observed in *B. byssoidea* MUCL94 compared to *B. pseudocinerea* VD165 and *B. cinerea* B05.10. On the other hand, during the infection in the *Raphanus sativus* leaves, induction of the *mfsG* gene in *B. cinerea* B05.10 was higher than in *B. pseudocinerea* VD165 and *B. byssoidea* MUCL94 (Figure 6).

Furthermore, all infections exhibited a similar trend in the level of *mfsG* gene induction in the three *Botrytis* species, except for the aforementioned exceptions. However, in the case of *Brassica oleracea* var. italica, the induction observed in the three *Botrytis* species was the lowest of all infections conducted (Figure 6).

## 3. Discussion

In this study, six *Botrytis* species were systematically evaluated for their tolerance to isothiocyanates, with a particular focus on the role of the *mfsG* gene. Of the *Botrytis* species studied (*B. cinerea*, *B. pseudocinerea*, *B. byssoidea*, *B. deweyae*, *B. fabae*, and *B. convolute*), three were identified as having the *mfsG* gene, while the *mfsG* gene was not detected in the remaining three by amplifying various fragments of the *mfsG* gene by PCR. The experimental design involved exposure to PITC, BITC, and PhITC at concentrations ranging from 0 to 1000 µM. In that way, the results shown in Figure 2 and Figure 4 indicate a clear link between the presence of the *mfsG* gene and an increase in tolerance to these isothiocyanate products (PITC, BITC, and PhITC), as it was reported by Vela-Corcía in 2019 with the generation of the knock-out mutant of the *mfsG* gene on *B. cinerea* B05.10, causing a dose-dependent inhibition in the presence of these isothiocyanates (PITC, BITC, and PhITC) on the knock-out mutant [23]. The growth of the species in which the *mfsG* gene was detected was less inhibited compared to those in which no *mfsG* gene was detected by PCR. Remarkably, *B. pseudocinerea* VD165 was the most resistant of the species in which the *mfsG* gene was detected by PCR, followed by *B. cinerea* B05.10, whereas *B. byssoidea* MUCL94 exhibited heightened sensitivity. In contrast, among the species with the *mfsG* gene not detected by PCR, *B. deweyae* exhibited elevated tolerance for all the isothiocyanates tested (PITC, BITC, and PhITC), while *B. convolute* was marginally more resistant than *B. fabae*, except for BITC, where the opposite was observed. These findings support recent evidence that suggests that our understanding of the glucosinolate breakdown mechanism in plants is incomplete, as is the list of antifungal agents. Previous research has concluded that unknown fungal decomposition products may contribute to in vivo antifungal effects [38,39,40]. Our results underscore the importance of fully understanding glucosinolate breakdown processes and their relation to plant antifungal resistance. Further research is clearly needed to explore the potential of some breakdown products that may contribute to the in vivo antifungal response and enhance the overall understanding of plant-fungus interactions [41].

Additionally, dose-dependent responses were evident across all *Botrytis* species, with BITC showing the highest toxicity, followed by PhITC (Table 2). PITC demonstrated the least toxicity, even at concentrations where growth inhibition was negligible in species in which the *mfsG* gene was not detected by PCR, such as *B. deweyae*. All these data open new avenues to explore the relevance of this gene in the tolerance to isothiocyanates in each of the species studied in this work, generating knock-out mutants in those species in which the *mfsG* gene was detected by PCR or complementing with the *mfsG* gene at the *niaD* locus in those *Botrytis* species in which the presence of the *mfsG* gene was not detected.

Subsequently, infectivity assays showed that the species in which the *mfsG* gene was detected by PCR could successfully infect *B. oleracea*, *R. sativus* leaves, *R. sativus* tubers, and *B. oleracea* var. italica vegetables. In that sense, the presence of the *mfsG* gene confirms tolerance to isothiocyanate products, acquiring an important role during infection [23]. However, this fact cannot be extrapolated to the whole Brassicaceae family (characterized by producing high levels of glucosinolates), as in the case of broccoli infections. In this case, *Botrytis* species, in which the *mfsG* gene was not detected by PCR, are the most pathogenic. On the other hand, infection of broccoli plants by *B. convolute* MUCLII595, *B. fabae* 2220, *B. deweyae* B1, *B. byssoidea* MUCL94, and *B. pseudocinerea* VD165 has been reported for the first time in this work (Figure 5). In that way, the results described by Vela-Corcía in 2019 [23] support the majority of our results, but they cannot be extrapolated to all pathogen–Brassicaceae species combinations, especially in the case of broccoli. This observation underscores the intricate nature of plant-pathogen interactions, necessitating a refined interpretation of genetic associations. In cabbage and radish leaves, as well as radish in tuber form, *B. pseudocinerea* consistently emerged as the most aggressive species among the species in which the *mfsG* gene was detected by PCR (Figure 5). However, in radish tubers, *B. cinerea* exhibited higher aggression. Interestingly, the *Botrytis* species in which the *mfsG* gene was not detected by PCR failed to infect these substrates in vitro. However, due to the large amount of GLs that are present in the *Brassica* family—at least 24 GSLs have been identified so far, including 16 aliphatic GSLs [progoitrin (PRO), sinigrin (SIN), glucoiberin (GIB), gluconapin (GNP), glucoraphanin (GRA), glucoiberverin (GIV), glucobrassicanapin (GBN), glucoraphasatin (GRH), edusaehanin, glucoerucin (GER), glucosativin (GST), glucoalyssin, gluconapoleiferin, glucocheirolin, glucoberteroin (GBT), and epiprogoitrin], 4 indolic GSLs [glucobrassicin (GBS), neoglucobrassincin (NGBS), 4-methoxyglucobr-assicin (4-MGBS), and 4-hydroxyglucobrassicin (4-OHGBS)], and 4 aromatic GSLs [gluconasturtiin (GNS), glucotropaeolin (GTP), glucobarbarin, and glucosinalbin] [35]—further analyses are needed to better understand the relationship of this gene with the type of compounds present in the Brassica family of plants. 

Previous studies have reported different GLS compositions in broccoli with respect to other Brassicaceae vegetable types [37]. So, among other less abundant GLSs, the analysis of broccoli revealed 65% of aliphatic GLSs, the majority GRA, and approximately 34% of indole GLSs (28% 4MGBS and 6% GBR). Although a lot of different factors, like solubility or polarity, might be at play, the observed differences in the aliphatic/indolic ratio with respect to other Brassicaceae species could explain the differential infective behavior in broccoli [36].

An expression analysis of the *mfsG* gene elucidated significant induction in the presence of isothiocyanates, both in axenic culture and after infection with Brassicaceae family plants. This induction was more than 60 times greater than the control levels, emphasizing the pivotal role of this gene in responding to these compounds. Alternate research has suggested dual roles for ABC and MFS transporters, potentially serving as effectors or contributors to the secretion of pathogenicity factors, influencing fungal virulence [42]. Dos Santos et al.’s research into MFS transporters in yeast proposed an additional function—providing drug resistance through indirect regulation of stress response and membrane potential control [26]. This prompts consideration of analogous impacts in plant-pathogen transporters, where similar multifaceted effects on virulence and resistance mechanisms may be at play [26]. These insights broaden our understanding of transporter functions, indicating their potential significance in both pathogenicity and drug resistance across diverse biological systems [27,43,44,45]. In that way, all the data obtained shed light on the significant impact of pure compounds—specifically, PITC, BITC, and PhITC—and the genetic response in different plants due to infection. Despite the diversity of the plant species studied, a consistent pattern emerged from the axenic cultures supplemented with isothiocyanates and from the infection experiments. In all the conditions and species evaluated, the mfsG gene was induced more than 60-fold over the control in axenic culture without isothiocyanate supplementation (Figure 6). This remarkable and uniform increase in gene expression suggests the robust and widespread influence of PITC, BITC, and PhITC on the genetic machinery of plants. This induction could be indicative of a sophisticated interaction between these isothiocyanate products and the genetic regulatory mechanisms of plants. These findings suggest the potential role of these compounds in enhancing plant defenses or triggering other vital physiological responses, despite the fact that the type and concentration of glucosinolates vary between cultivars [35]. However, this study opens intriguing avenues for further research, inviting exploration into the specific molecular pathways affected and the broader implications for plant health, stress responses, and defense mechanisms in the context of these isothiocyanate products.

In addition, the conservation analysis of amino acids within the catalytic center proposed by Vela-Corcía in 2019 [23] demonstrated notable conservation across diverse *Botrytis* species, suggesting the potential significance of these amino acids in protein activity. In that way, bioinformatic exploration provides valuable insights into the evolutionary conservation of key amino acids among the *Botrytis* species studied. In conclusion, the consistent presence and conservation of these amino acids across different species implies functional relevance, highlighting their potential role in the activity of the protein under investigation [46].

On the other hand, apart from the mfsG gene, other detoxification mechanisms in different fungal species, such as Sclerotinia sclerotium, have been described [25]. In that sense, it has been reported that, during infection of Brassicaceae plants, S. sclerotium cleverly manipulates the glucosinolate/myrosinase system by metabolizing isothiocyanates through two pathways [25]. While one involves conjugation to glutathione, the fungus excels in a second, much more efficient pathway utilizing an isothiocyanate hydrolase analogous to a bacterial enzyme [25]. This enzyme transforms isothiocyanates into non-toxic compounds, promoting fungus growth despite the plants’ defenses. This adaptation contributes significantly to the virulence of *S. sclerotiorum* on glucosinolate-producing plants, revealing a sophisticated fungal strategy to propagate and thrive amidst the biochemical complexities of plant-fungal interactions [25].

This comprehensive inquiry provides valuable insights into the dynamics between *Botrytis* species and isothiocyanates, underscoring the vital role of the *mfsG* gene in conferring tolerance. These findings call for continued research in this field, advocating the use of advanced biotechnological tools to unravel the intricate biology of these proteins and deepen our understanding of plant-pathogen interactions. The findings also generate heightened interest in the use of diverse biotechnological tools to deepen our understanding of the biology surrounding this family of transporters. Biotechnological approaches such as genomic tools can aid in unraveling the genetic architecture, identifying regulatory elements, and understanding the broader genetic networks associated with these proteins [47]. On the other hand, proteomic techniques can provide insights into the protein’s structure, interactions, and potential post-translational modifications [48,49]. In addition, CRISPR-Cas9 gene editing technology can enable targeted modifications, allowing researchers to validate the functional significance of specific genes and amino acids identified in their studies [50,51]. Unraveling the biology of this family of proteins can inform the development of targeted strategies to enhance plant resistance, optimize crop yield, and mitigate the impact of plant diseases. In conclusion, the multifaceted data presented in this report advocate for a sustained and diversified research approach, leveraging biotechnological tools to decipher the biology of this family of transporters. 

## 4. Materials and Methods

### 4.1. Bioinformatic Analysis

The nucleotide sequence of the *mfsG* gene was used to search for homologous genes in other *Botrytis* species using the BLASTn tool [52,53] and the nucleotide collection database (nr/nt). For species that are poorly sequenced and for which there is no gene databank, the Whole-genome shotgun contigs (wgs) database was used. The domain was analyzed with the Pfam v34.0 database from the Conserve Domain Database (CDD) search tool from the National Center for Biotechnology Information (NCBI) [54,55] using the *mfsG* gene coding sequence. To analyze the amino acid conservation of the MfsG protein sequence, a blastp was used to find homologous proteins in other *Botrytis* species using the non-redundant protein sequences (nr) database [53]. The protein sequences from *B. cinerea* B05.10, *B. fragariae* BVB16, and *B. byssoidea* MUCL94 were aligned with the structural alignment (Expresso) from the T-coffe tool [56], https://tcoffee.crg.eu/apps/tcoffee/references.html (accessed on 10 January 2024). The results were visualized with the Espript 3.0 tool [57] “https://espript.ibcp.fr (accessed on 10 January 2024)”. The conserved amino acids proposed by Vela-Corcía et al. in 2019 [23] to be the binding region (Trp54, Phe254, Gln344, and Phe369), as well as the amino acids from *B. cinerea* B05.10, *B. byssoidea* MUCL94, and *B. fragariae* BVB16 involved in the hydrogen binding of BITC and PITC (Trp54) and PhITC (Arg290), were displayed in the alignment. On the other hand, well-conserved nucleotide alignment regions displayed in the alignment were used for primer design aimed at the detection of the *mfsG* gene with PCR analysis (Appendix A). The primers used for this proposal amplify different regions conserved among the *Botrytis* species (including *B. cinerea* B05.10, *B. pseudocinerea* BP362, *B. medusae* B555, *B. fragariae* BCB16, and *B. byssoidea* MUCL94) within the major facilitator superfamily (pfam07690; interval 136–1140).

All the data from the genomes of the different species of *Botrytis* used in this bioinformatic approach are available in the NCBI database with the following Genebank accession numbers: *B. cinerea* B05.10-GCA_000143535.4, *B. pseudocinerea* BP362-GCA_019395245.1, *B. medusae* B555-GCA_019395255.1, *B. fragariae* BCB16-GCA_013461495.1, *B. byssoidea* MUCL92-GCA_014898295.1, *Botrytis fabae* G12-GCA_032594075.1, *Botrytis deweyae* B1-GCA_014898535.1, *Botrytis squamosa* MUCL 31421-GCA_014898485.2, *Botrytis sinoallii* Bc 23-GCA_014898435.1, *Botrytis aclada* 633-GCA_014898285.1, *Botrytis galanthina* MUCL435-GCA_004916875.1, *Botrytis elliptica* Be9401-GCA_024478385.1, *Botrytis porri* MUCL3432-GCA_014898465.1, *Botrytis hyacinthi* Bh0001-GCA_004786245.1, *Botrytis tulipae* Bt9001-GCA_004786125.1, *Botrytis paeoniae* Bp0003-GCA_004786145.1, *Botryotinia globosa* MUCL444-GCA_014898425.1, *Botryotinia narcissicola* MUCL2120-GCA_004786225.1, *Botryotinia convolute* MUCL11595-GCA_004786275.1, and *Botryotinia calthae* MUCL2830- GCA_004379285.1. All species have genome coverage higher than 50.0×, except *B. edusae* (20.0×), *B. calthae* (17.0×), *B. pseudocinerea* (20.0×), *B. narcissicola* (19.0×), *B. galanthina* (35.0×), and *B. porri* (42.0×).

### 4.2. Organisms, Media, and Culture Conditions

The *Botrytis* species used in this study were *B. cinerea* B05.10 [30,31], *B. pseudocinerea* VD165 [58], *B. deweyae* B1 [59], *B. byssoidea* MUCL94 [60], *B. fabae* 2220, and *B. convolute* MUCLII595 [61]. All fungal cultures underwent standard cultivation in a YGG medium (composed of 2% glucose, 0.5% yeast extract, 0.3% Gamborg’s B5 medium from Duchefa Biochemie, and, when necessary, 1.5% agar) at a temperature of 20 °C for three days. Conidia were produced on tomato agar plates (comprising 25% homogenized tomato fruit by weight and 1.5% agar; pH 5.5). These conidial stocks were preserved in a 10% glycerol solution at −80 °C. For non-sporulate species, mycelium plugs were preserved in a 20% and 60% glycerol stock solution at −80 °C. 

*B. oleracea* and *R. sativus* leaves, *R. sativus* tubers, and *B. oleracea* var. italica vegetables were purchased from local groceries. All the plants used in this study were selected according to their morphology, condition, and apparent stage.

### 4.3. Standard Molecular Method for mfsG Gene Detection

Traditional molecular methodologies were implemented for gene manipulation. Fungal genomic DNA extraction adhered to the established protocol outlined by Mansfield (1985) [62]. DNA integrity and concentration were assessed via spectrophotometric analysis using a NanoDrop 2000c instrument (Thermo Scientific, Waltham, MA, USA). PCR reactions were executed employing Phusion High-Fidelity DNA Polymerase (Thermo Scientific) and Go-Taq DNA Polymerase (Promega, Madison, WI, USA). Sequence analysis was conducted utilizing software from the DNASTAR Lasergene package (DNASTAR, Inc., Madison, WI, USA), and oligonucleotide primers were sourced from Metabion International AG. Primers for PCR detection of the *mfsG* gene are provided in Appendix A.

### 4.4. Vegetative Growth

To examine the vegetative growth inhibition of the fungi in the presence of different isothiocyanate products, Petri dishes were prepared with YGG-agar medium and subsequently inoculated with three mycelial agar plugs measuring 3 mm in diameter. Incubation was at 20 °C, and any deviation in fungal growth patterns was monitored. Daily measurements of the colony radius were recorded over a period of three days. The assessment of the radial growth rate involved plotting the colony radius against time and fitting the data into a linear model, which displayed a robust correlation (Pearson’s correlation coefficient value (r^2^ ≥ 0.98)). To evaluate the strains’ responsiveness to stress stimuli, the YGG-agar medium was supplemented with different concentrations (5 µM, 10 µM, 20 µM, 50 µM, 100 µM, 200 µM, 500 µM, and 1000 µM) of isothiocyanate products (PITC (ref.305310250; Thermo Scientific Acros), BITC (ref.252492; Sigma-Aldrich, Burlington, MA, USA), and PhITC (ref.8070280025; Sigma-Aldrich)) to induce stress caused by the presence of plant defense products, which are isothiocyanate products. All the concentrations tested were dissolved in 1 mL of MeOH, with the control involving the addition of 1 mL of MeOH without any ITC. All the plates were prepared and used the same day due to the volatility of some of the isothiocyanates employed in this work. In that way, the isothiocyanates were added once the media were tempered to 40 °C after the sterilization process. The radial mycelial growth radius (R, in mm) was measured at 72 h for the treatments with the isothiocyanate products (Rt) and the controls without any supplementation (Rc) in YGG medium. The fungitoxic effects of PITC, BITC, and PhITC were quantified as the percentage of growth inhibition (GI) using the following formula: GI (%) = [(Rc − Rt)/Rc] × 100. The results were represented as the means of the growth inhibition percentages of 50 biological replicates (*n* = 50) distributed across 5 independent experiments, each consisting of 10 biological replicates, presented alongside their respective standard deviations. To calculate the half-maximal concentration (IC50), the inhibition data were taken for each of the concentrations evaluated for each of the *Botrytis* species tested that conformed to a logarithmic trend (r^2^ > 0.98). From the logarithmic equation for each species, the IC50 value was found.

### 4.5. Virulence Assay

Virulence assessments were conducted using detached *B. oleracea* and *R. sativus* leaves, *R. sativus* tubers, and *B. oleracea* var. italica vegetables. These substrates were inoculated with agar plugs measuring 3 mm in diameter in a 10 µL TGGK solution (composed of 60 mM KH_2_PO_4_, 10 mM glycine, 0.01% Tween 20, and 100 mM glucose). Incubation ensued in darkness at 20 °C, maintaining high humidity levels, with photographic documentation every 24 h. ImageJ software v1.8.0 [63] (developed at the U.S. agency National Institutes of Health) facilitated the measurement of lesion diameter on *B. oleracea* and *R. sativus* leaves, *R. sativus* tubers, and *B. oleracea* var. italica vegetables. Results are presented as the mean lesion diameters of 50 biological replicates, derived from 2 independent experiments with 25 biological replicates each, from the infection of the different *Botrytis* species after 3 days of infection. In the case of leaves, a total of 50 leaves were used. Each leaf was inoculated with all 6 species. In the case of fruit, due to their size, 300 fruits were used (50 fruits for each of the 6 species). Each individual fruit was inoculated with one of the species.

### 4.6. Quantitative Real-Time PCR (qRT-PCR)

Total RNA extraction from fungal mycelia entailed a filtration process from 30 mL of YGG medium (control) or YGG supplemented with 80 µM of PITC, BITC, and PhITC inoculated with 5 agar plugs measuring 0.3 cm in diameter. This culture was left to incubate at 20 °C in darkness for three days. To explore gene expression within plant hosts, detached *B. oleracea* and *R. sativus* leaves, *R. sativus* tubers, and *B. oleracea* var. italica vegetables were inoculated with 5 µL droplets containing a mycelial plug measuring 0.3 cm in diameter in a 10 µL TGGK solution composed of 60 mM KH_2_PO_4_, 10 mM glycine, 0.01% Tween 20, and 100 mM glucose. All inoculated species were incubated for 72 h at 22 °C in a dark environment, maintaining a humidity level of 70%. After infection, segments of the affected leaf areas were excised and promptly frozen at −80 °C for subsequent analysis. Both mycelia and infected plant tissues were homogenized, and total RNA was isolated using Trizol Reagent (Sigma-Aldrich, T9424) following the manufacturer’s protocol.

The subsequent step Involved cDNA synthesis from 1 µg of total RNA using the iScript cDNA Synthesis Kit (Bio-Rad, Hercules, CA, USA), following the instructions provided. Quantitative real-time PCR (qRT-PCR) was conducted on an iCycler iQ system (Bio-Rad, USA) using the iQ SYBR Green Supermix (Bio-Rad, USA) along with the primers detailed in Appendix A. To ensure normalization of the cDNA samples, the actA (Bcin16g02020) and B-tub (Bcin01g08040) genes, responsible for encoding actin and tubulin, respectively, were used as internal controls. Relative mRNA quantities were determined using the ΔΔCt method [64], averaging six biological replicates (*n* = 6), each containing six technical determinations of the threshold cycle (Ct). The resulting data were presented as mean values of $2^{\left(\Delta \Delta Ct\pm SD\right)}$ normalized with respect to the expression of each gene in YGG without the addition of the PITC, BITC, and PhITC isothiocyanates.

Furthermore, the relative expression of the *mfsG* gene was measured in the species containing the indicated transporter gene (*B. cinerea* B05.10, *B. pseudocinerea* VD165, and *B. byssoidea* MUCL94). This study was carried out in axenic culture after 72 h of incubation with the PITC, BITC, and PhITC isothiocyanates at a single concentration of 70 µM and after 72 h of infection on Brassicaceae family plants. The axenic culture of these *Botrytis* species after 72 h of incubation in YGG medium without the addition of any other compound was used as a control. This control was used to normalize the data and ascertain the extent to which this gene was induced under each condition compared to the control.

### 4.7. Statistical Analysis

Graphpad Prism 8 was used for statistical analyses. The Kolmogorov-Smirnov test (for samples > 50) and the Shapiro-Wilk test (for samples < 50) were used to assess normality. Depending on normality outcomes, the *t*-test or Mann-Whitney test was applied to compare normally distributed or nonparametric data, respectively. Statistical significance was established at a *p*-value < 0.05.

## 5. Conclusions

In this study, six *Botrytis* species were systematically studied for their response to isothiocyanates, particularly focusing on the *mfsG* gene. The presence of this gene was associated with increased tolerance to isothiocyanate compounds, as demonstrated by reduced growth inhibition percentages in mfsG-containing species. However, the correlation between the *mfsG* gene and tolerance was challenged, particularly in the case of broccoli, suggesting the complex nature of plant-pathogen interactions.

Concentration-dependent responses were observed across all *Botrytis* species, with benzyl isothiocyanate (BITC) exhibiting the highest toxicity. Infectivity assays revealed varying aggressiveness among species on different plant substrates, challenging previous findings and highlighting the need for a nuanced interpretation of genetic associations.

The expression analysis of the *mfsG* gene showed significant induction in the presence of isothiocyanates, emphasizing its pivotal role in responding to these compounds. Bioinformatic investigations identified homologous *mfsG* genes in specific *Botrytis* species, with conserved amino acids in the catalytic center suggesting their functional significance.

This study underscores the importance of advanced biotechnological tools, including genomics, proteomics, bioinformatics, and CRISPR-Cas9 gene editing, in deciphering the intricate biology of these proteins. This comprehensive approach not only enhances our understanding of plant-pathogen interactions but also holds promise for developing targeted strategies to improve plant resistance and optimize crop yield. In conclusion, continued and diversified research using advanced biotechnological tools is essential for unlocking the full potential of these proteins and offering innovative solutions for agriculture and crop management.

## Figures and Tables

**Figure 1 plants-13-00756-f001:**
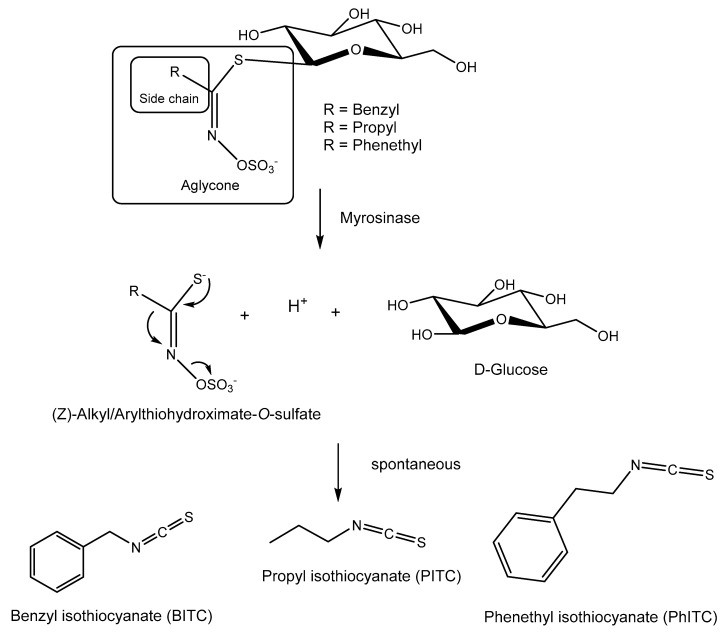
Glucosinolates (GLSs) and their enzymatic and spontaneous conversion to BITC, PITC, and PhITC.

**Figure 2 plants-13-00756-f002:**
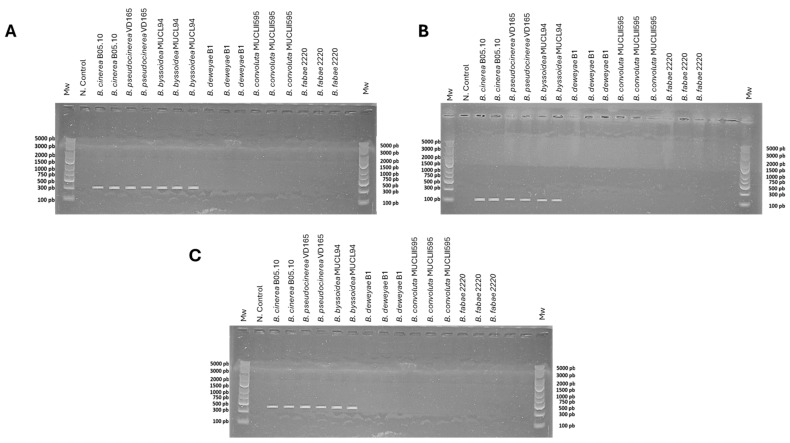
PCR analysis for the detection of the *mfsG* gene in *B. cinerea* B05.10, *B. pseudocinerea* VD165, *B. byssoidea* MUCL94, *B. deweyae* B1, *B. fabae* 2220, and *B. convolute* MUCLII595 using primer pairs (**A**) CKmfsG1/CKmfsG2 (335 bp), (**B**) CKmfsG3/CKmfsG4 (114 bp), and (**C**) CKmfsG1/CKmfsG4 (478 bp). The negative control was replacing genomic DNA with water. For the species in which the *mfsG* gene was bioinformatically detected (*B. cinerea*, *B. pseudocinerea*, and *B. byssoidea*), two or more PCRs from various independent genomic DNA extractions were performed, while for the species in which *mfsG* was not bioinformatically detected (*B. deweyae*, *B. fabae*, and *B. convolute*), three PCRs from three independent genomic DNA extractions were carried out. Mw: molecular weight (O’GeneRuler Express DNA Ladder, ready to use from Thermofisher Scientific, Madrid, Spain; ref. SM1563).

**Figure 3 plants-13-00756-f003:**
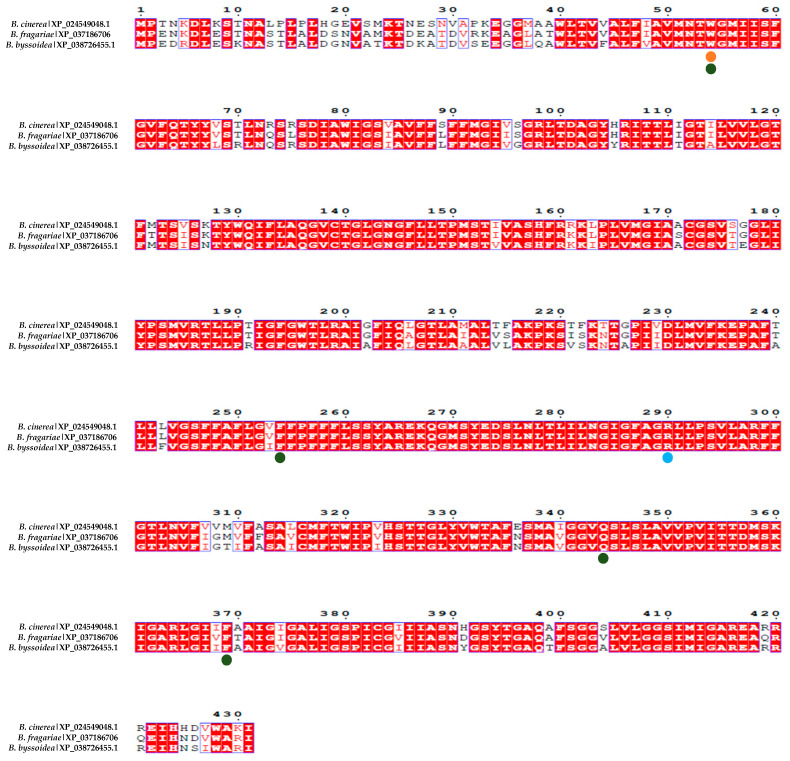
Alignment of the MfsG homologous protein in other species of the *Botrytis* genus. The alignment in red represents highly conserved amino acids among the different MfsG sequences. XP_024549048.1 represents the MfsG protein sequence from *B. cinerea*, XP_037186706.1 represents the MfsG protein sequence from *B. pseudocinerea*, and XP_038726455.1 represents the MfsG protein sequence from *B. byssoidea*. The green dots represent the conservation of the proposed amino acids involved in the binding region (Trp 54, Phe254, Gln344, and Phe369); the yellow dots represent the predicted amino acids involved in the hydrogen binding to PITC and BITC; and the blue dots are for the predicted amino acids involved in the PhITC hydrogen binding previously described in [23].

**Figure 4 plants-13-00756-f004:**
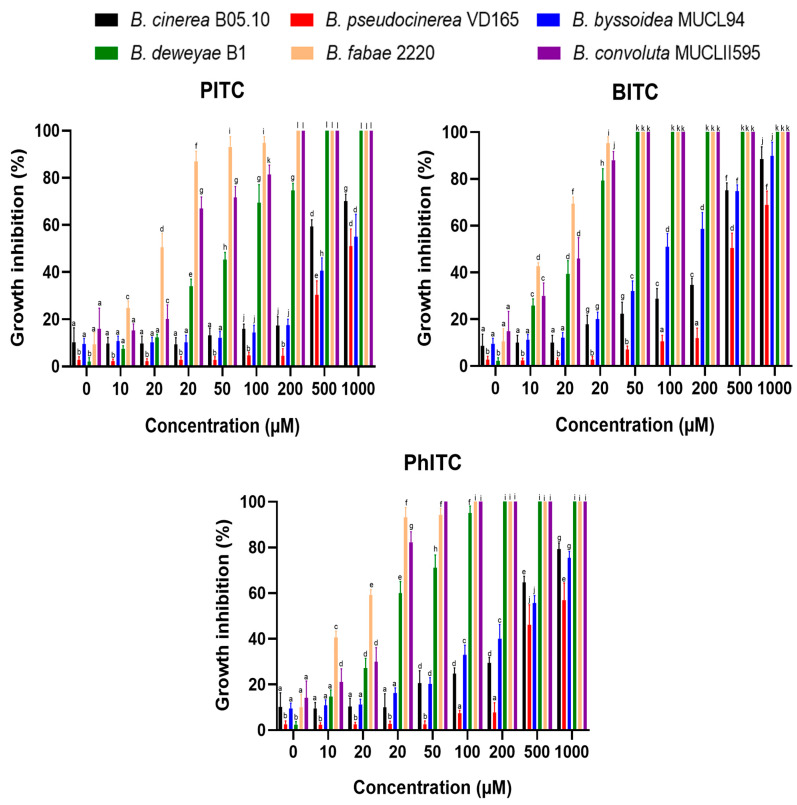
Percentage inhibition of *B. cinerea* B05.10, *B. pseudocinerea* VD165, *B. byssoidea* MUCL94, *B. deweyae* B1, *B. fabae* 2220, and *B. convolute* MUCLII595 to different isothiocyanates (PITC, BITC, and PhITC). The data shown are the means of 50 biological replicates (*n* = 50) distributed across 5 independent experiments, each consisting of 10 biological replicates. The different letters above each bar indicate significant differences (*p* < 0.05) between the different *Botrytis* species analyzed.

**Figure 5 plants-13-00756-f005:**
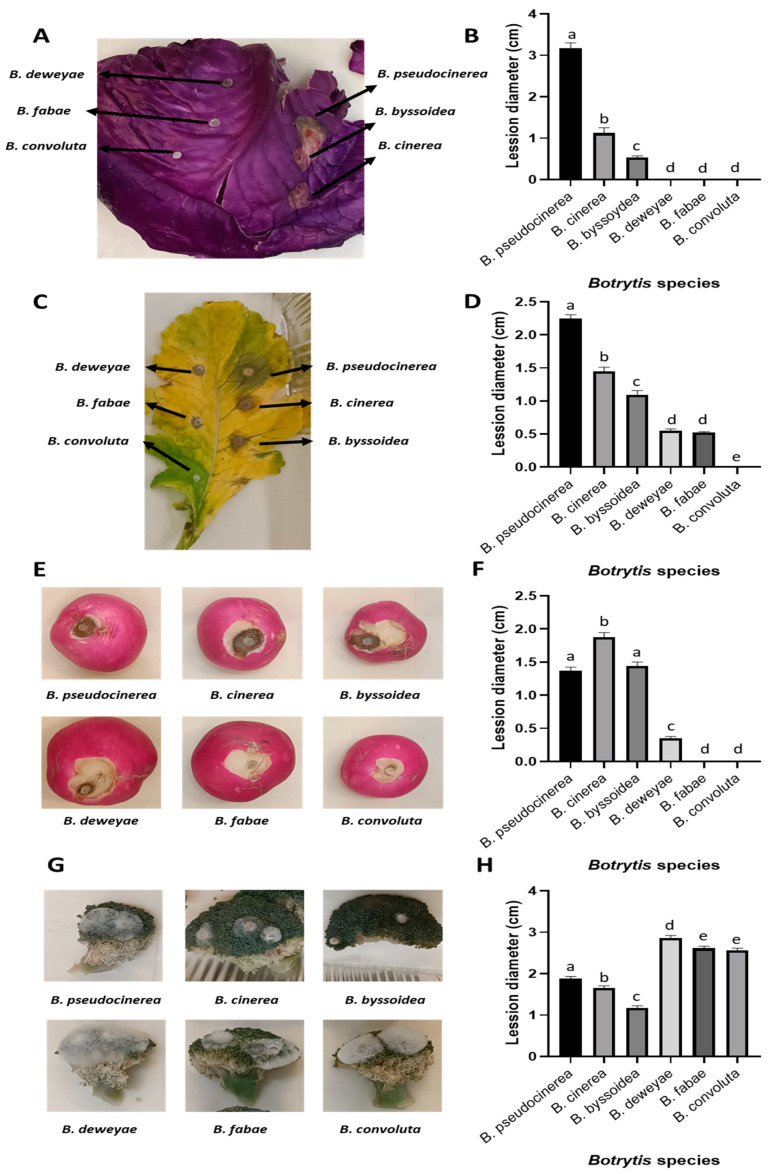
Infection assay of different members of the Brassicaceae family inoculated with *B. cinerea*, *B. pseudocinerea*, *B. byssoidea*, *B. deweyae*, *B. fabae*, and *B. convolute*. Representative images of the lesions produced by each fungal strain after 3 days in (**A**) *B. oleracea*, (**C**) *R. sativus* leaves, (**E**) *R. sativus* tubers, and (**G**) *B. oleracea* var. italica vegetables. Lesion diameter after 3 days of infection with the different *Botrytis* species in (**B**) *B. oleracea*, (**D**) *R. sativus* leaves, (**F**) *R. sativus* tubers, and (**H**) *B. oleracea* var. italica vegetables. The data shown in (**B**,**D**,**F**,**H**) correspond to the means of 50 biological replicates (*n* = 50), derived from 2 independent experiments with 25 biological replicates each. The different letters above each bar indicate significant differences (*p* < 0.05) between the different *Botrytis* species analyzed.

**Figure 6 plants-13-00756-f006:**
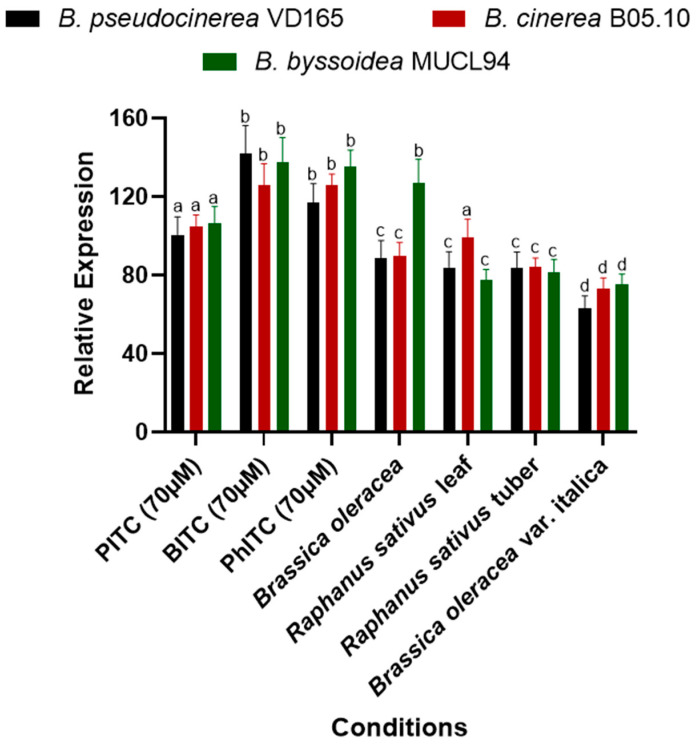
Relative expression of the *mfsG* gene of *B. cinerea*, *B. pseudocinerea*, and *B. byssoidea* in reaction to the presence of thiocyanates in axenic culture (PITC, BITC, and PhITC) after 72 h of growth and during infection in high-glucosinolate (Brassicaceae family) vegetables such as *B. oleracea* and *R. sativus* leaves, *R. sativus* tubers, and *B. oleracea* var. italica vegetables after 72 h of inoculation. Data are plotted as relative expression compared to the expression level of the different *Botrytis* species after 72 h of growth on axenic culture without any supplementation (YGG-agar medium), set as 1. The different letters above each bar indicate significant differences (*p* < 0.05) between the different *Botrytis* species analyzed. The data shown correspond to the mean values of 6 biological replicates (*n* = 6) performed using 6 technical replicates for each biological replicate.

**Table 1 plants-13-00756-t001:** Bioinformatic detection of the *mfsG* gene among different *Botrytis* species.

Species	Isolate	NCBI Gene Accession Number	NCBI Protein Accession Number	Gene Identity with *B. cinerea* B05.10 (%)	Length (pb)	Reference
*B. cinerea*	B05.10	XM_024693262.1	XP_024549048.1	100%	1379	[30,31]
*B. fragariae*	BVB16	XM_037342786.1	XP_037186706.1	86.72%	1296	[32]
*B. byssoidea*	MUCL 94	XM_038882714.1	XP_038726455.1	84.47%	1296	[33]
*B. pseudocinerea*	BP362	JAHXJK010000103.1: 18998-20539	-	99.33%	1541	[34]
*B. medusae*	B555	JAHXJK010000103.1: 15498-16733	-	90.47%	1235	[34]

**Table 2 plants-13-00756-t002:** IC50 values of different *Botrytis* species for PITC, BITC, and PhITC treatments.

*Botrytis* Species	Types of Isothiocyanates		
	PITC	BITC	PhITC
*B. cinerea* B05.10	469.96 µM	288.99 µM	361.981 µM
*B. pseudocinerea* VD165	971.65 µM	552.51 µM	707.04 µM
*B. byssoidea* MUCL94	786.22 µM	131.92 µM	334.84 µM
*B. deweyae* B1	49.21 µM	10.82 µM	19.85 µM
*B. fabae* 2220	9.56 µM	6.04 µM	6.87 µM
*B. convolute* MUCLII595	19.62 µM	9.15 µM	11.98 µM

## Data Availability

Data are contained within the article and Appendix A.

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
