# Peer review of "Comparing Fungal Sensitivity to Isothiocyanate Products on Different Botrytis spp."

_plants, 2024, doi:10.3390/plants13060756_

Round 1

Reviewer 1 Report

Comments and Suggestions for Authors

Dear Authors,

In my opinion, some issues regarding presentation of data should be carried out before the paper is suitable for publication. Despite the above, my opinion is positive overall. Please consult my suggestions below.

Major issues.

1., The difference of sensitivities of the two groups should be submitted to statistical inference. The statistical power is low (n = 3 per group), but still, talking about "difference" requires some sort of test to be carried out.

The true significance ("%" contribution to pathogenicity) of the mfsG transporter gene could be drawn from experiments using unmodified, knockout and recovered strains. The current setup has the limitation that several other factors are changed between various species.

Modify the discussion accordingly.

2., How do we know for sure that the given isolates do not contain mfsG? Provide data on this. Were these not tested or tested, but not detected (Fig. 4. and elsewhere)? Or, were there genome sequences used to check for this?

3a., Section 4.3.: Where were the vegetables from?

3b., Fig. 4.: The photos do not show seedlings, but developed organs.

4., Were all plates prepared fresh and used for inoculation on the same day? This is relevant for PITC which is quite volatile.

5., The discussion repeats the results a bit too often. However, I missed some discussion on the following topics: (1) I would speculate that perhaps the differences in the indolic - aliphatic GSL ratio might be behind the differential behavior in broccoli. Different pathogens can induce the increased biosyntheis of different GSL classes. Obviously a lot of other factors might be in play. (2) What do we know of the efficacy of mfsG products on various ITCs? Might it contribute to the different potency of the tested ITCs? Or, are we seeing a gradient along a rank in polarity?

Minor issues

L92-93 and Fig. 2.: There is a lack of an equally spaced dose-response curve: the tested range is too much for the sensitive strains and not potent enough against the tolerant ones. If you also tested 5, 10, 500 and 1000, IC50 values might have been calcualted. I agree though that the current data might sepearate strains to two groups.

Fig.1.: PhITC compound structure is wrong. The correct one is PhCH2CH2NCS.

L115: were more successful

L128: I do not think this is an in vitro setup

L267: I'd say the findings of ref. 30 cannot be extrapolated to all pathogen - Brassicaceae species combinations.

L245: You did not conduct comparative tests on other decomposition products, so your study does not support this claim. Either remove or cite relevant literature.

Best regards.

Author Response

Dear Reviewer 1,

We sincerely appreciate your insightful comments and suggestions regarding our manuscript. Following your suggestions, we hope that in general we have improved the presentation of the results.

1., The difference of sensitivities of the two groups should be submitted to statistical inference. The statistical power is low (n = 3 per group), but still, talking about "difference" requires some sort of test to be carried out.

 The true significance ("%" contribution to pathogenicity) of the mfsG transporter gene could be drawn from experiments using unmodified, knockout and recovered strains. The current setup has the limitation that several other factors are changed between various species.

 Modify the discussion accordingly.

Following your suggestions, the number of concentrations to be tested was increased between 0-1000 µM. The concentrations evaluated for each compound were 0, 5, 10, 20, 50, 100, 200, 500 and 1000 µM and the percentage inhibition was calculated with respect to the axenic culture medium for each of the compounds tested. In this way, statistical tests could be performed to verify the difference in sensitivity.  The figure caption of Figure 4 (lines 210-214) has been modified. In addition, half maximal inhibitory concentration (IC50) was calculated (lines 496-499). The discussion was modified accordingly to the introduced changes. So, the results of the IC50 calculations that are shown in table 2 are explained in lines 181-193.

 We agree with your comment about using unmodified and knockout transformants. Works are in progress to obtain the indicated mutants. So, we have included a paragraph in Discussion where we indicate that “All these data open new avenues to explore the relevance of this gene in the tolerance to glucosinolate degradation products in each of the species studied, with null mutants in those species in which this gene was detected by PCR or complementation at the niaD locus of the mfsG gene in those strains in which the presence of this gene was not detected”, lines 322-326.

2., How do we know for sure that the given isolates do not contain mfsG? Provide data on this. Were these not tested or tested, but not detected (Fig. 4. and elsewhere)? Or, were there genome sequences used to check for this?

To provide experimental evidence about the detection or non-detection of the mfsG gene in other Botrytis species, a diagnostic PCR (lines 463-471) was performed by choosing conserved regions within the nucleotide sequences of the species identified by the blastn search (Supplementary Figure 1 and lines 426-431), using 2 pairs of primers for this purpose, having the option to amplify 3 different regions. The results are presented from line 98 to 114 and figure 2.

3a., Section 4.3.: Where were the vegetables from?

Vegetables were purchased from local groceries. All the plants used in this study were selected according to their morphology, condition, and apparent stage. See, section 4.2, lines 458-460.

3b., Fig. 4.: The photos do not show seedlings, but developed organs.

We have changed seedling for vegetable in the legend of figure 4, in this version figure 5, lines 273-281.

 4., Were all plates prepared fresh and used for inoculation on the same day? This is relevant for PITC which is quite volatile.

Yes, effectively all plates were prepared fresh and used for inoculation on the same day. See section 4.4 , lines 487-488.

5., The discussion repeats the results a bit too often. However, I missed some discussion on the following topics: (1) I would speculate that perhaps the differences in the indolic - aliphatic GSL ratio might be behind the differential behavior in broccoli. Different pathogens can induce the increased biosyntheis of different GSL classes. Obviously a lot of other factors might be in play. (2) What do we know of the efficacy of mfsG products on various ITCs? Might it contribute to the different potency of the tested ITCs? Or, are we seeing a gradient along a rank in polarity?

We have introduced a paragraph speculating about differences in the composition of GLS, in broccoli, respect other Brassicaceae vegetables, and the differential  indolic- aliphatic GSL ratio in broccoli.

“Previous studies have reported the different composition of GLS, in broccoli, with respect to other Brassicaceae vegetables type. So, among others minority GLS, the analysis of broccoli revealed 65 % of aliphatic GLS and approximately 34% of indole GLS (28% (4MGBS), and 6% (GBR)).  Although a lot of different factors like solubility or polarity, might be in play, the observed differences in the aliphatic-indolic ratio with respect other Brassicaceae species could explain the differential infective behavior in broccoli.” Lines 358-364.

Minor issues:

L92-93 and Fig. 2.: There is a lack of an equally spaced dose-response curve: the tested range is too much for the sensitive strains and not potent enough against the tolerant ones. If you also tested 5, 10, 500 and 1000, IC50 values might have been calculated. I agree though that the current data might separate strains to two groups.

Attending your recommendations, the suggested concentrations were included, and the half maximal inhibitory concentration (IC50) was calculated (lines 496-499). Changes are added in lines 181-193 to add the results of the IC50 calculations that are shown in table 2, lines 205-206. Figure 4 has been changed and corrected.

Fig.1.: PhITC compound structure is wrong. The correct one is PhCH2CH2NCS. Figure 1 was corrected

Line 115, now line 224 in the new manuscript, was corrected

Line 128, now line 228, “in vitro” was removed

Line 267, now line 339-340, it was corrected saying “that it can not be extrapolated to all pathogen - Brassicaceae species combinations

Line 244, now line 315,:  The sentence, “our data unequivocally reinforce that isothiocyanates (ITCs) are the most potent anti-fungal decomposition products of glucosinolates” was eliminated.

We hope you will find the revised version of our manuscript engaging and an improved reflection of our research intentions.

Yours sincerely,

Reviewer 2 Report

Comments and Suggestions for Authors

The manuscript by Coca-Ruiz et al. is aimed at testing a possible correlation between presence of the gene encoding Major Facilitator Superfamily transporter G (msfG) in six Botrytis species, their tolerance to glucosinolate-derived isothiocyanates and their virulence to vegetables of the genus Brassica. The topic is interesting, but the manuscript is incomplete /premature and, unfortunately, it is a quite sloppy representation of the experimental work. Due to missing or improperly presented information, I am not able to really judge about the core data and the conclusions that have been drawn by the authors. My detailed comments are listed below.

1. The introduction is to the most part a copy of the introduction of the paper by Vela-Corcia et al. (2019) Nature Communications. Some single words have been changed and a few sentences been added, but this kind of taking over a text generated by others without pointing out that this text has been taken (more or less as an entity) from another source, is unacceptable. According to generally accepted rules of good scientific practice, this is a case of plagiarism.

2. The authors claim that they studied three Botrytis species with and three without the msfG gene. But they don’t provide data in support of a lack of this gene in three species other than a table stating that the gene is lacking. Apart from BLAST searches, the authors should apply some additional methods, especially to the poorly sequenced Botrytis species or those with no GenBank entry. They should state clearly, for which species a completely annotated genome is available and for which species complete genome information is lacking. From the present manuscript, one cannot exclude the possibility that the msfG gene was not found because of incompleteness of the genome sequence and/or its annotation. Moreover, the authors should provide at least some experimental support for presence/absence of the gene in the six species.

3. As the assignment as msfG-containing/non-containing species is kind of a starting point for the rest of this work as it is presented in the manuscript, this assignment should be the first section of the results part (not the last).

4. The methods part lacks information on which plant cultivars from which source were used and how plants were cultivated and when/how material was harvested.

5. The source of the chemicals (ITCs) is lacking. How were poorly water-soluble ITCs dissolved to obtain the given concentrations in the medium?

6. The formula for calculating growth inhibition in % is wrong (or at least unclear). Should be % GI = (Rc-Rt)/(Rc*100)

7. qPCR: the same house-keeping genes should be used for each of the species; presently it does not look like it based on Table S1.

8. Throughout all the experimental work it needs to be stated clearly (in methods and figure /table legends), how many technical replicates were done for one biological sample, how many biological replicates were included per experiment, and how many independent experiments were conducted (and are these presented all together or is one experiment presented that has been repeated with the same rpincipal result). Technical replicates should be used to calculate the mean for one replicate. The means of all replicates then go into calculation of the biological result with variation (mean +/- SD, and N is the number of biological replicates).

9. As cultivars of Brassica species (as well as organs and developmental stages) vary a lot in their glucosinolate content (e.g. there are mild and very hot radishes), the virulence tests are only meaningful in the context of the present study, if glucosinolate levels are determined. Moreover, there are about 120 different glucosinolate structures known to date. The vegetables that were tested produce different glucosinolates than those from which the tested ITCs derive. There are various types of breakdown products in planta, not just ITCs. For example, broccoli expresses an ESP and produces epithionitrile instead of ITC. Chemical composition of the tested plant material is therefore very critical for interpretation of the data.

10. Line 116/117: This reference did not test different species of the genus Arabidopsis, but different genotypes of Arabidopsis thaliana (i.e. various mutants).

11. Please provide qPCR raw data data as suppl. material/excel sheets.

12. The title of the paper (and of section 2.1) should refer to isothiocyanates (or glucosinolate-derived isothiocyanates), other breakdown products were not tested.

13. Several parts of the results section do not belong there, but rather to the methods or discussion section.

Figure 1: Structure of 2-phenylethyl-ITC is wrong. Correct “myrosinasa” to -ase.

Figure 2: Assignment of species as with/without msfG gene does not match the assignment in Table 1.

Table 1: B. fabae is missing.

Table S1: Primers named Fw/RvACT are being used to quantify tubulin gene expression? – something has been mixed up here.

Table S2: It is not very meaningful to present a table with statistical output that is not being used to present the data in the main text. Moreover, it is not clear which data are actually referred to here: % growth inhibition or colony diameter. Such a table only provides useful information, if it shows kind of the raw data (colony diameter) and if it becomes clear how many technical replicates and biological replicates have been analyzed. Statistical measures that need to be provided are: N (sample size), result of normality check, appropriate measure (mean and/or median) with variation (SD and/or min/max). This should match the figures in the main text.

Comments on the Quality of English Language

Meaning of some sentences or phrases is unclear, as it is not always obvious to which previously used word "this"/"these"/"there" refers.

Author Response

Dear Reviewer 2,

We sincerely appreciate your insightful comments and suggestions regarding our manuscript. Following your suggestions, we hope that in general we have improved the presentation of the manuscript.

The introduction is to the most part a copy of the introduction of the paper by Vela-Corcia et al. (2019) Nature Communications. Some single words have been changed and a few sentences been added, but this kind of taking over a text generated by others without pointing out that this text has been taken (more or less as an entity) from another source, is unacceptable. According to generally accepted rules of good scientific practice, this is a case of plagiarism.

First of all, we would like to apologize for the similarity of some paragraphs in the introduction of our article with those of Vela-Corsia et al.,. This fact has been due to the lack of experience of our young student who prepared the draft of the introduction. The student was impressed by the excellent work of Vela-Corsia, et al. and he based his introduction on said article considering the importance of the results described. We have taken action in this regard, and we want to express our most sincere apologies.

The introduction has been completely rewritten and we want to emphasize that the experimental reported in our article submitted to Plants journal is totally novel and that none of the experiments included in the article have been previously reported.

  1. The authors claim that they studied three Botrytis species with and three without the msfG gene. But they don’t provide data in support of a lack of this gene in three species other than a table stating that the gene is lacking. Apart from BLAST searches, the authors should apply some additional methods, especially to the poorly sequenced Botrytis species or those with no GenBank entry. They should state clearly, for which species a completely annotated genome is available and for which species complete genome information is lacking. From the present manuscript, one cannot exclude the possibility that the msfG gene was not found because of incompleteness of the genome sequence and/or its annotation. Moreover, the authors should provide at least some experimental support for presence/absence of the gene in the six species.

To provide experimental evidence about the detection or non-detection of the mfsG gene in other Botrytis species, a diagnostic PCR (lines 463-471) was performed by choosing conserved regions within the nucleotide sequences of the species identified by the blastn search (Supplementary Figure 1 and lines 426-446), using 2 pairs of primers for this purpose, having the option to amplify 3 different regions. The results are presented from line 98 to 103 and figure 2.

For the bioinformatics analysis, detection was performed based on the taxid botrytis/botryotinia in the blastp and blastn. This 2 taxid comprises 20 species other than botrytis with genome coverage greater than 50.0x except for B. meduase (20.0x), B. calthae (17.0x), B. pseudocinerea (20.0x), B. narcissicola (19.0x), B. galanthina (35.0x) and B. porri (42.0x) (lines 432-446). However, with the experimental evidence performed by PCR, fragments belonging to this gene are detected in B. pseudocinerea. In lines 434-437 are provided the reference of the genome from the Botrytis species in which the mfsG gene was detected by blastp. In addition, as you can appreciated in the material and method section, some of the strains used in the experimental study (listed on table 2) are different from the reference strains whose sequencing is published (used for the bioinformatic study and listed in table 1). In this sense, the study is carried out together with the PCR confirmation of 3 species in which the gene is bioinformatically detected and 3 species in which it is not detected, corresponding to the strains available in the culture collection of our department.

  1. As the assignment as msfG-containing/non-containing species is kind of a starting point for the rest of this work as it is presented in the manuscript, this assignment should be the first section of the results part (not the last).

Following the recommendation about the bioinformatics study, it has become part of the first results section starting on line 86 of the text (reordering the rest of the figures, tables and results sections).

  1. The methods part lacks information on which plant cultivars from which source were used and how plants were cultivated and when/how material was harvested.
  2. oleracea and R. sativus leaves, R. sativus tuber and B. oleracea var. italica vegetable were purchased from local groceries. All the plants used in this study were selected according to their morphology, condition and apparent stage (lines 458-460)
  3. The source of the chemicals (ITCs) is lacking. How were poorly water-soluble ITCs dissolved to obtain the given concentrations in the medium?

From line 482-484 are listed the references of the ITCs products. All the concentrations tested were dissolved in 1mL of MeOH being the control the addition of 1 mL of MeOH without any ITC, lines 485-488.

  1. The formula for calculating growth inhibition in % is wrong (or at least unclear). Should be % GI = (Rc-Rt)/(Rc*100)

The formula used has been changed to a better comprehension of it to GI(%) = [(Rc—Rt)/Rc)] × 100 in line 493, but it gives the same results as the one previously written. This formula is the same as Vela-Corcía et,al. [23] uses in his article to calculate the growth inhibition.

  1. qPCR: the same house-keeping genes should be used for each of the species; presently it does not look like it based on Table S1.

The same house-keeping genes ActinA and β-tubulin are used for each of the species. It is written in the description of the primers in table S1. There are one pair of primers for the mfsG gene, actA gene and β-tubulin (the two last one, corresponding to the house-keeping gene) for each specie.

  1. Throughout all the experimental work it needs to be stated clearly (in methods and figure /table legends), how many technical replicates were done for one biological sample, how many biological replicates were included per experiment, and how many independent experiments were conducted (and are these presented all together or is one experiment presented that has been repeated with the same rpincipal result). Technical replicates should be used to calculate the mean for one replicate. The means of all replicates then go into calculation of the biological result with variation (mean +/- SD, and N is the number of biological replicates).

Following your suggestion, the number of biological replicates, number of technical replicates and the independent experiments that were performed were clarified in the material and methods section and in all the figure titles. (lines 212-213, 278-279, 290-291, 494-495, 509-510 and 535-536)

  1. As cultivars of Brassica species (as well as organs and developmental stages) vary a lot in their glucosinolate content (e.g. there are mild and very hot radishes), the virulence tests are only meaningful in the context of the present study, if glucosinolate levels are determined. Moreover, there are about 120 different glucosinolate structures known to date. The vegetables that were tested produce different glucosinolates than those from which the tested ITCs derive. There are various types of breakdown products in planta, not just ITCs. For example, broccoli expresses an ESP and produces epithionitrile instead of ITC. Chemical composition of the tested plant material is therefore very critical for interpretation of the data.

We know that the content of ITCs varies according to each of the plant materials used (lines 347-357). The objective of this work was not to analyze the composition of all the ITCs present in each of the plants used, but to evaluate the tolerance or degree of infectivity of these two groups of fungi (those containing the mfsG gene and those not containing the mfsG gene detected by PCR) to different types of plants that present ITCs. Your proposal, it is very interesting, which we will take into consideration for future works.

  1. Line 116/117: This reference did not test different species of the genus Arabidopsis, but different genotypes of Arabidopsis thaliana (i.e. various mutants).

In line 226-227 it was changed different species of Arabidopsis to different genotypes of Arabidopsis.

  1. Please provide qPCR raw data data as suppl. material/excel sheets.

A supplementary excel sheet with raw data is provided.

  1. The title of the paper (and of section 2.1) should refer to isothiocyanates (or glucosinolate-derived isothiocyanates), other breakdown products were not tested.

According to the suggestion, the title was changed including glucosinolate-derived isothiocyanates and section 2.1 that now is 2.3 in line 168-169.

  1. Several parts of the results section do not belong there, but rather to the methods or discussion section.

Following your recommendation, we have carefully revised both sections. In our opinion, perhaps certain phrases that could be in other sections, are emphasized in the results section, to clarify the results to the readers.

Figure 1: Structure of 2-phenylethyl-ITC is wrong. Correct “myrosinasa” to -ase.

The structure of 2-phenylethyl-ITC was corrected and the myrosinasa was corrected to myrosinase in Figure 1.

Figure 2: Assignment of species as with/without msfG gene does not match the assignment in Table 1.

Species detected by blastn or blastp are the strains listed in table 1 that whose genome are available at the ncbi. The strains assignment of species as with/without msfG gene are the strains available in our laboratory that was used for the PCR analysis and for the experimental part of this paper. Some strains used on the experimental part (lines 449-451) are not the same strains whose genome are available on the NCBI database (line 432-446).

Table 1: B. fabae is missing. In table 1

  1. fabae is not missing due to it is the result of the bioinformatic analysis. They are not the strains used in the experimental part, section 4.2, lines 429-431.

Table S1: Primers named Fw/RvACT are being used to quantify tubulin gene expression? – something has been mixed up here.

In table S1, the description of the primers Fw/RvACT were corrected because it was for the actin gene expression quantification.

Table S2: It is not very meaningful to present a table with statistical output that is not being used to present the data in the main text. Moreover, it is not clear which data are actually referred to here: % growth inhibition or colony diameter. Such a table only provides useful information, if it shows kind of the raw data (colony diameter) and if it becomes clear how many technical replicates and biological replicates have been analyzed. Statistical measures that need to be provided are: N (sample size), result of normality check, appropriate measure (mean and/or median) with variation (SD and/or min/max). This should match the figures in the main text.

According to the suggestion, the statistical table was eliminated from this section.

We truly appreciate the effort and time that the reviewer put into the evaluation of our article. Their insights have been invaluable in strengthening our paper.

Yours sincerely,

Reviewer 3 Report

Comments and Suggestions for Authors

In this study, six Botrytis species were systematically studied for their response to isothiocyanates. The authors found that the presence of mfsG was associated with increased tolerance to isothiocyanate compounds, and reported first that B. fabae, B. deweyae, B. byssoidea and B. pseudocinerea could infect broccoli.

In Figure 3. Which hosts were used in A, C, E, or G, need to be described one by one, instead of vaguely saying that there are four members of the Brassicaceae family: B. oleracea and R. sativus leaves, R. sativus tuber and B. oleracea var. italica vegetable were assayed in this figure.

In Figure 4, relative expression calculation compared to which treatment? Please add details of control treatment in the figure legend. Significance analysis should be added.

“2.2. Glucosinolate breakdown products and infection processes induce the mfsG gene”。This section should refer to Figure 4, which is not yet cited in the text.

Author Response

Response reviewer 3

Dear Reviewer:

We would like to express our gratitude for your valuable comments on our manuscript.

In Figure 3. Which hosts were used in A, C, E, or G, need to be described one by one, instead of vaguely saying that there are four members of the Brassicaceae family: B. oleracea and R. sativus leaves, R. sativus tuber and B. oleracea var. italica vegetable were assayed in this figure.

In this new version, following the comments of referee 2, we have changed the bioinformatic analysis to the section 2.1, then Figure 3 is now Figure 5. Following your right suggestions we have indicated the host one by one in A, C, E and G, as well as, in lesion diameter B, D, F and H. (see legend figure 5, see lines 274-278)

In Figure 4, relative expression calculation compared to which treatment? Please add details of control treatment in the figure legend. Significance analysis should be added.

 Following your valuable suggestions, Figure 4, in the new version Figure 6, we have introduced relative expression compared to the expression level on axenic culture without any supplementation of the different botrytis species after 72 h of growth. Significance analysis has been added (lines 287-291)

“2.2. Glucosinolate breakdown products and infection processes induce the mfsG gene”。This section should refer to Figure 4, which is not yet cited in the text.

In the section 2.2, now section 2.4, we have referred the figure 4, now figure 6 (line 256).

We are immensely grateful for the reviewer's comments.

Yours sincerely,

Round 2

Reviewer 2 Report

Comments and Suggestions for Authors

The manuscript by Coca-Ruiz et al. has been revised and improved. But there are still some important points that need attention:

L 40 correct to “in the presence of specifier proteins that convert the glucosinolate aglucone to other products” (these proteins do not act on the ITC, but on the aglucone)

L65-80 Please check thoroughly if the citations are correct. My impression is that something has been mixed up here. At least the text does not seem to be a very good and suitable representation of the cited studies. e.g. L68: ref. 24

L68-69 What do you mean with this sentence? Please specify about which species you talk here or about which other MFS transporters

L84 Correct to isothiocyanates

L84 and throughout the text: refer to the compounds as “isothiocyanates” (prefered) or as “isothiocyanate products”

L94 remove doubling (“but in both cases”)

L122 and following: what do you mean with hydrogen binding of BITC. Please use propper and precise terms.

L126-135: remove, this belongs to the discussion

L170-171: mention the compounds tested here with ref. to figure 1

L171-174: rephrase, pay attention to the word order

L174-178: remove, this information describes method details (which are also obvious from the figures)

L179: it is not necessary to refer to fig. 2 here, please refer to fig. 4 for the results

L186-187: Please state that you only refer to the comparison between B. cinerea and B. B. byssoides here (otherwise the statement would not be correct)

L189: what is a PCR gene?

L199: not detected

L215: start a new section here (2.4) as you come to very different experiments here. (2.4 then becomes 2.5 and so on)

L220: Please mention (with ref.) that glucosinolate content may vary between cultivars.

L226-232: remove, belongs to the discussion; moreover, how can you make a comparison between tests on broccoli and tests on Arabidopsis? For all other vegetable the results were as expected. I would not say that your results contradict those of Vela-Corcia. it is just an unexpected result that raises questions about broccoli as a substrate.

L247-268: Please present the results of the expression analysis first (what was done in principle, with which result). The present text belongs to the discussion and needs to be removed here. Please remember (for the discussion) that you have not analyzed the type or amount of various breakdown products in the plants. You simply assume that the breakdown products would be ITCs, but this does not have to be the case. If a plant like broccoli produces nitriles instead, you cannot expect effects of ITCs.

Please go through the discussion critically and thoroughly, but also concise and appropriate with respect to the data presented.

Table 2: Please add the units for the concentrations.

Fig. 2: There is a band in A for the first replicate of B. deweyae. This needs to be reported in the text and discussed.

Fig. 3: Add species names to the alignment

Suppl. Fig. 1: indicate specifically the positions of the used primers for detection of the MFS transporter gene (with name/abbreviation of primer)

Comments on the Quality of English Language

see above, needs improvement

Author Response

Dear Reviewer 2,

We sincerely appreciate your insightful comments and suggestions regarding our manuscript. Following your suggestions, we hope that in general we have improved the presentation of the manuscript.

L 40 correct to “in the presence of specifier proteins that convert the glucosinolate aglucone to other products” (these proteins do not act on the ITC, but on the aglucone)

According to the suggestion, line 40 was corrected saying “in the presence of specifier proteins that convert the glucosinolate aglucone to other products”.

L65-80 Please check thoroughly if the citations are correct. My impression is that something has been mixed up here. At least the text does not seem to be a very good and suitable representation of the cited studies. e.g. L68: ref. 24

Certainly, and thanks to your comment, the references from line 65-80, now 68-84 was completely revised and reorganized.

L68-69 What do you mean with this sentence? Please specify about which species you talk here or about which other MFS transporters

The lines 68-72, now 68-76 was reorganized and changed from: “Furthermore, additional studies have identified the MFS transporter crucial for the detoxification of isothiocyanates (ITCs), the breakdown products of GLSs [25]. This transporter exhibits differential expression patterns in response to various ITCs, with heightened expression levels observed during interactions with wild-type A. thaliana [26,27],” to “In addition, this transporter exhibits differential expression patterns in response to various ITCs, with heightened expression levels observed during interactions with wild-type A. thaliana [23]. Furthermore, additional studies have identified others detoxification mechanisms such as the one present in Sclerotinia sclerotiorum that effectively decomposed ITCs via enzymatic hydrolysis, with SsSaxA protein playing a crucial role on it. However, a mutant lacking the SssaxA gene was unable to thrive in host plants producing glucosinolates [25]. Moreover, other studies have emphasized the importance of this family of transporters (MFS) being required for example for multi-drug/multixenobiotic resistance in yeast (Saccharomyces cerevisiae) [26].”

L84 Correct to isothiocyanates

In line 84, now line 88, it was corrected the compound name to isothiocyanates

L84 and throughout the text: refer to the compounds as “isothiocyanates” (prefered) or as “isothiocyanate products”

According with your tips, all the text was revised and changed when we were talking about the compound in terms of Glucosinolate-derived isothiocyanate products for your suggestion to name the compound as isothiocyanates or isothiocyanate products in line 2, line 88, line 127, line 167, line 178, line 213, line 233, line 235, line 316, line 496 and line 548

L94 remove doubling (“but in both cases”)

In line 94, now line 98 it was removed doubling “but in both cases”

L122 and following: what do you mean with hydrogen binding of BITC. Please use propper and precise terms.

In line 122 now line 126, the phrase “amino acids involved in the hydrogen binding of BITC and PITC (Trp54) and PhITC (Arg290)” was rewritten according to your suggestion for a better understanding with the appropriate terms “amino acids predicted to bind to the isothiocyanates BITC and PITC (Trp54) and PhITC (Arg290) via hydrogen bonds [23]”

L126-135: remove, this belongs to the discussion

Following to your recommendation, previous lines 126-135, were moved to the discussion in line 378-383

L170-171: mention the compounds tested here with ref. to figure 1

In line 170-171, now 167-168, the three compounds PITC, BITC and PhITC were mentioned in the text and referred to reference 1.

L171-174: rephrase, pay attention to the word order

Line 171-174, now 168-173 were rephrase avoiding repetition of content. The new paragraph is: “In three of these six species (B. cinerea B05.10, B. pseudocinerea VD165 and B. byssoidea MUCL94) the mfsG gene was detected amplifying by PCR different fragments of the gene, while in the other three species (B. deweyae B1, B. fabae 2220 and B. convolute MU-CLII595) the mfsG gene was not detected with the amplification of different fragments of the gene by PCR.”

L174-178: remove, this information describes method details (which are also obvious from the figures)

According to the suggestions, previous lines 174-178 were moved from the results to the discussion section in lines 384-389. The phrase was simplified to “In that way bioinformatic exploration provides valuable insights into the evolutionary conservation of key amino acids among the Botrytis species studied. In conclusion, the consistent presence and conservation of these amino acids across different species implies functional relevance, highlighting their potential role in the activity of the protein under investigation [46]”

L179: it is not necessary to refer to fig. 2 here, please refer to fig. 4 for the results

According to your suggestion figure 4 was referred now in line 176 and the reference to figure 2 (in previous line 179, now line 176) was removed.

L186-187: Please state that you only refer to the comparison between B. cinerea and B. B. byssoides here (otherwise the statement would not be correct)

Following your recommendation, line 186-187, now 181-183 was rewritten as “B. cinerea B05.10 shows a higher IC50 value than B. byssoidea MUCL94, indicating a higher degree of resistance to BITC and PhITC (Table 2 and Figure 4) than B. cinerea B05.10”.

L189: what is a PCR gene?

Line 189, now line 184-185, it was corrected writing: “On the other hand, from the species which the mfsG gene was not detected by PCR…”.

L199: not detected

In line 199, now line 193 “no detected” was changed to “not detected”.

L215: start a new section here (2.4) as you come to very different experiments here. (2.4 then becomes 2.5 and so on)

According to your suggestion, in line 215, now lines 210 and 211 it was added the section 2.4, called: “Infection assays of Botrytis species with the mfsG gene detected and not detected, on different Brassicaceae species.” Previous section 2.4 now is section 2.5 in lines 233-234.

L220: Please mention (with ref.) that glucosinolate content may vary between cultivars.

Following your recommendation, in line 220, now lines 216-217, it was mentioned: “In addition, previous studies reported that the glucosinolate production varies between cultivars [36,37]”

L226-232: remove, belongs to the discussion; moreover, how can you make a comparison between tests on broccoli and tests on Arabidopsis? For all other vegetable the results were as expected. I would not say that your results contradict those of Vela-Corcia. it is just an unexpected result that raises questions about broccoli as a substrate.

Following to your recommendation, previous lines 226-232 was moved to the discussion and rewritten in lines 321-328: “In that sense, the presence of the mfsG gene confirms tolerance to isothiocyanate products, acquiring an important role during infection [23]. However, this fact cannot be extrapolated to the whole Brassicaceae family (characterized by producing high levels of glucosinolates) as in the case of broccoli infections. In this case, Botrytis species which the mfsG gene was not detected by PCR, are the most pathogenic. On the other hand, infection of broccoli plants by B. convolute MUCLII595, B. fabae 2220, B. deweyae B1, B. byssoidea MUCL94 and B. pseudocinerea VD165 is reported in this work for first time (Figure 5).”

L247-268: Please present the results of the expression analysis first (what was done in principle, with which result). The present text belongs to the discussion and needs to be removed here. Please remember (for the discussion) that you have not analyzed the type or amount of various breakdown products in the plants. You simply assume that the breakdown products would be ITCs, but this does not have to be the case. If a plant like broccoli produces nitriles instead, you cannot expect effects of ITCs.

Regarding to the suggestion, the section 2.5 was rewritten (lines 235-259) trying to show only the results and replacing part of what was written in to the discussion section (lines 366-381).

Please go through the discussion critically and thoroughly, but also concise and appropriate with respect to the data presented.

Discussion of results was analyzed and revised exhaustively

Table 2: Please add the units for the concentrations.

The units for the concentration were added to Table 2.

Fig. 2: There is a band in A for the first replicate of B. deweyae. This needs to be reported in the text and discussed.

It was an error for this gel labelling the PCR lanes. In the case of A, the mistake was that 3 PCR was performed for B. byssoidea and I named B. deweyae the last lane from B. byssoidea. It was changed also in the text now in line 115 “the 2 PCR from 2 independent genomic DNA extraction” for “two or more PCRs from various independent genomic DNA extractions” due to B. byssoidea in A, there are 3 replicates.

Fig. 3: Add species names to the alignment

Following your recommendation species name were added to the alignment in Figure 3.

Suppl. Fig. 1: indicate specifically the positions of the used primers for detection of the MFS transporter gene (with name/abbreviation of primer)

The position of the primers for the detection of the mfsG gene was displayed with arrows in Supplementary Figure 1. On the title of Suppl. Fig. 1, the following phrase was added: “The arrows indicate the orientation of the primers used for the mfsG gene detection.”

We truly appreciate the effort and time that the reviewer put into the evaluation of our article. Their insights have been invaluable in strengthening our paper.

Yours sincerely,
